# Comparison between Early Clinical Results of Dual-Linear and Conventional Foot-Pedal Control in Phacoemulsification

**DOI:** 10.3390/jcm13030693

**Published:** 2024-01-25

**Authors:** Hyungil Kim, Jiyun Seong, Changrae Rho

**Affiliations:** 1Gyeongju St. Mary’s Eye Clinic, Gyeongju 38146, Republic of Korea; lasikdoctor@naver.com; 2Saevit Eye Hospital, Goyang 10447, Republic of Korea; jyseong04040@gmail.com

**Keywords:** cataract, phacoemulsification, corneal endothelium, phaco machine

## Abstract

Background: The aim of this study was to compare early clinical results regarding the safety and efficacy of dual-linear vs. conventional foot-pedal control in cataract surgery. Methods: This was a paired-eye contralateral, retrospective, observational study. Each patient underwent cataract surgery in both eyes: one eye with dual-linear foot-pedal control (study group) and the other eye with conventional foot-pedal control (control group). Absolute phaco time (APT), average phaco power, effective phaco time (EPT), and surgical complications were analyzed and compared. Corneal endothelial cell count, corneal thickness, corneal volume, and best-corrected distance visual acuity (BCDVA) were measured preoperatively and at 1 week, 1 month, and 3 months postoperatively. Results: A total of 94 patients (188 eyes) were enrolled. The respective APT, average phaco power, and EPT values were 7.05 ± 5.31 s, 28.4 ± 1.00, and 2.05 ± 1.56 s in the study group and 6.82 ± 6.48 s, 18.9 ± 1.74, and 1.35 ± 1.35 s in the control group. Conclusions: The average phaco power and EPT values were significantly higher in the study group. The safety of the dual-linear foot pedal was comparable to that of a conventional pedal in terms of endothelial cell loss, central corneal thickness, and surgical complications.

## 1. Introduction

Senile or presenile cataracts are the leading causes of visual impairment and cataract surgery is the most widely performed ocular procedure in most countries [1,2,3]. The surgical volume is expected to increase as populations age [2]. The rate of cataract surgery increased from 13.4 per 1000 Medicare beneficiaries in 1980 to 58.7 per 1000 beneficiaries in 1995 [2]. The field of cataract surgery is one of the most representative areas reflecting cutting-edge technology. State-of-the-art technology is applied to all stages of surgery, including preoperative biometric measurements, intraoperative surgical instruments, intraocular lenses, intraocular lens formulae, the phacoemulsification machine, and postoperative care, including dry eye [3,4,5,6,7,8,9,10,11]. Mechanical and instrumental advancements have shortened the surgical time. The increased accuracy of biometric measurements and lens formulae greatly reduced the postoperative refractive error. Improvements in the intraocular lenses made it possible to correct astigmatism and to reverse presbyopia. Nonetheless, in the era of refractive cataract surgery, improved results and a reduction in ocular damage remain important goals [12].

Phacoemulsification machines have evolved to include a torsional ultrasound delivery system or femtosecond laser to deliver ultrasound energy efficiently or decrease the total ultrasound energy [13,14,15]. Another recent advancement has been the switch to dual-linear foot-pedal control (Stellaris Elite^®^, Bausch+Lomb, St. Louis, MO, USA). Compared to pitch-only conventional foot-pedal control, the new system has two different planes of pedal movement: the conventional plane (pitch) allowing ultrasound control and a new lateral movement (yaw) that powers the vacuum control, or vice versa. With dual-linear foot- pedal control, the surgeon can modulate the vacuum independent of the phaco energy, allowing the increased vacuum to facilitate movement of the nucleus towards the phaco tip in order to hold and chop the nuclear fragment. Thereafter, low to moderate vacuum pressure can be applied in nuclear emulsification.

The overall three-dimensional control provided by the dual-linear foot-pedal system should improve the surgical procedure, as the ultrasound energy is delivered to the eye more efficiently. The use of low to moderate vacuum pressure in emulsification improves safety by decreasing the risk of inadvertently damaging the posterior lens capsule. The Stellaris Elite includes adaptive fluidics, which allows active monitoring of the vacuum flow rate and automatic adjustment of the infusion pressure.

We compared the early clinical results achieved using the dual-linear foot-pedal control vs. conventional foot pedal in terms of the safety and efficacy of cataract surgery.

## 2. Materials and Methods

### 2.1. Patients

This was a paired-eye contralateral, retrospective, observational study. The study was approved by the Institutional Review Board of Gyeongju St. Mary’s Eye Clinic (GSM-2023-05, 1 May 2023) and its protocol followed the principles of the Declaration of Helsinki. Informed consent was obtained from each patient.

The inclusion criteria were patients >45 years old with cataracts in both eyes who underwent phacoemulsification between 1 March 2022 and 28 February 2023, using conventional foot-pedal control in one eye and dual-linear control in the contralateral eye. The first operated eye was randomly assigned to either the dual-linear- or conventional foot-pedal control group using a randomization card. The severity of cataracts was graded according to the Lens Opacities Classification System III (LOCS III) by the same physician. Eyes with nuclear opacity classified between 2+ and 6+ were included in this study. The exclusion criteria were a history of ocular trauma, previous ocular surgery, a history of serious ocular comorbidities, other corneal disorders, active infection, uncontrolled glaucoma, optic atrophy, ocular tumor, known zonular weakness, or an endothelial cell count (ECC) < 1500 cells/mm^2^. Patients who missed their regular routine postoperative visits were also excluded.

### 2.2. Preoperative and Postoperative Assessment

All patients received a full ophthalmic examination that included best-corrected distance visual acuity (BCDVA), manifest refraction, slit-lamp biomicroscopic examination, noncontact intraocular pressure (IOP) measurement (BT-800, Topcon, Tokyo, Japan), ECC by a non-contact specular microscope (SP-1P, Topcon, Tokyo, Japan), central corneal thickness (CCT) and corneal volume (CV) by rotating Scheimpflug corneal tomography (Pentacam HR, Oculus Optikgeräte GmbH, Wetzlar, Germany), two kinds of Swept Source OCTs (IOLMaster 700, Carl Zeiss Meditec AG, Jena, Germany and Anterion, Heidelberg Engineering Ltd., Heidelberg, Germany) and fundus examination.

Postoperative evaluations included BCDVA, noncontact IOP, ECC, CCT, and CV at 1 week, 1 month, and 3 months after surgery. ECCs were calculated automatically by the embedded software. The percentage of endothelial cell loss (ECL) was calculated by dividing the pre- and postoperative endothelial cell difference by the preoperative ECC: ECL (%) = (ECCpre − ECCpost)/ECCpre. The change in CCT was obtained as follows: Change in CCT (%) = (CCTpost − CCTpre)/CCTpre. Pre stands for pre-operative and post stands for post-operative.

### 2.3. Surgical Procedure

All surgeries were performed by one experienced surgeon (Hyungil Kim), who has performed >25,000 cataract surgeries. The surgeon positioned himself at a superior aspect to the patient. The surgeon was ambidextrous. During surgeries performed on the right eye, the phaco handpiece was manipulated with the right hand and for the left eye the phaco handpiece was manipulated with the left hand.

Following application of topical anesthesia, cataract surgeries were performed using a standard phacoemulsification procedure with a 2.0-mm clear corneal incision made along the steepest axis (2.0 mm UniBevel Slit Knife, Angled, UniqueEdge, Sinking Spring, PA, USA) with the aid of a markerless alignment system (CALLISTO eye^®^, Carl Zeiss Meditec AG, Jena, Germany). The step axis was determined using Toric IOL summary at 4 mm obtained from Refractive Power/Corneal Analyzer (OPD-Scan III, NIDEK, Gamagori, Japan). The astigmatic magnitude was determined using total corneal power acquired from OPD-scan III, two swept source OCTs (Anterion and IOLMaster 700), and Dual Scheimpflug Tomography (Galilei G4, Ziemer Ophthalmology, Port, Switzerland). In the case where astigmatism is inconsistent and less than 0.5 D, the main incision was located at the superior temporal sclera to minimize surgically induced astigmatism. Side port incision was made 90° apart from the main incision.

After the anterior chamber had been filled by injection with sodium hyaluronate 1.5% (Hyalsan inj. prefilled, Daewoo pharm, Seoul, Republic of Korea), a 5.5–5.8-mm continuous curvilinear capsulorhexis was created using capsulorhexis forceps. Then, a cortical cleaving hydrodissection was performed to allow the nucleus to rotate freely in the capsular bag. A 23-gauge dual port cannula (AE-7655, HI KIM-Inamura cannula, Corzamedical Ophthalmology, Parsippany, NJ, USA) was used to inject a balanced salt solution in two separate directions to decrease the zonule stress. The nucleus was divided into four or more fragments by applying a pre-chop maneuver using a narrow, modified 1.7-mm-neck pre-chopper (AE-4298, HI Kim-Inamura pre-chopper, Corzamedical Ophthalmology.). For cataracts with a harder nucleus (>grade 4), the nucleus was fragmented via the counter pre-chop technique using both the pre-chopper and a universal chopper, with its rounded tip (AE-2591, HI Kim Chopper, Corzamedical Ophthalmology), to minimize possible zonule stress [16].

A Stellaris Elite^®^ platform was adopted for phacoemulsification using a BL3318S purple straight needle (outer diameter 0.74–0.94 mm, flared) and a BL3118 MICS sleeve. The ultrasound mode was set to the multiple burst phaco mode with a 30% amplitude, 75 burst duration, and six duty cycles. Two ultrasound foot-control settings were compared. For the control group, only the pitch mode was adopted, in which phacoemulsification started at position 3 after a full vacuum had been attained. For the study group (dual-linear foot control), pitch and yaw modulation, controlling the phaco power and the vacuum level, respectively, were used. A gravity infusion system, with an infusion bottle height of 70–80 cm, was used for the control group and adaptive fluidics, with a bottle height of 65 cm, for the study group.

After removal of the nucleus and cortex, the anterior chamber and capsular bag were reformed using an ophthalmic viscoelastic device (OVD) and a hydrophobic or hydrophilic intraocular lens was implanted. The determination of the intraocular lens power was mainly based on Anterion, with additional reference to IOLMaster 700. Depending on the type of lens, the incision width was enlarged to 2.2–2.5 mm. The OVD was exchanged for a balanced salt solution and the main wound was hydrated.

The efficacy of the operation using different foot-pedal settings was determined by comparing the absolute phaco time (APT), average phaco power, and effective phaco time (EPT). Safety was evaluated based on endothelial cell loss, CCT, CV, and surgical complications.

### 2.4. Statistics

The statistical analysis was performed using SPSS software (version 19.0, SPSS, Inc., Chicago, IL, USA). Continuous variables are presented as the mean and standard deviation, and categorical variables are given as a number. A *t* test was used to compare the two groups with respect to APT, average phaco power, EPT, ECL, CCT, and CV at 3 months postoperatively. An χ2 test was used to compare the nuclear-density grades. A *p* value < 0.05 was considered to indicate statistical significance.

## 3. Results

Of the 150 patients who underwent bilateral cataract surgery with different foot-pedal controls, 188 eyes from 94 patients (53% male) met the inclusion criteria. Table 1 shows the patients’ demographic and clinical data. The mean age was 68.9 ± 8.1 years (range: 49–89 years). The numbers of right eyes included in the study and control groups were 43 and 51, respectively. Differences in the baseline demographic data of the two groups were not statistically significant. Likewise, neither the differences in nuclear severity in each group assessed according to the LOCS III grade distribution nor the differences in the preoperative baseline BCDVA, ECC, CCT, and CV values were statistically significant. Biometric measurements such as anterior chamber depth, lens thickness, and axial length also showed no differences between the study and control groups.

The surgical results of the two groups are shown in Table 2. While APTs were comparable, the average phaco power was 50% larger in the study group than in the control group. Consequently, EPTs were statistically larger in the study group.

The postoperative surgical outcomes at 3 months are shown in Table 3. Corneal ECLs were 12.5 ± 11.0% in the study group and 14.8 ± 12.1% in the control group; the difference was not statistically significant. The absolute ECCs during the pre- and postoperative periods are shown in Table 4. The differences between the groups were not statistically significant (*p* > 0.05).

The absolute CCT measurements during the pre- and postoperative periods are shown in Table 5. The mean relative change (compared to baseline) in the CCT was 2.2 ± 3.4% at 1 week, 0.9 ± 2.0% at 1 month, and 0.4 ± 1.6% at 3 months in the study group and 1.6 ± 2.2% at 1 week, 0.6 ± 2.3% at 1 month, and 0.2 ± 2.2% at 3 months in the control group.

The respective surgical outcomes according to the nuclear opacity grade are summarized in Table 6. A subgroup analysis showed a statistically significant difference in average phaco-power values across all subgroups. The relative average phaco-power values in the study group compared to the control group were 1.49, 1.55, 1.49, 1.51, and 1.42 at NO2, NO3, NO4, NO5, and NO6, respectively. For all subgroups, ECLs were numerically lower in the study group than in the control group whereas the CCT and CV results were comparable.

### Complications

Overall, no significant surgery-related complications such as corneal-wound leakage, uncontrolled intraocular pressure, posterior capsular rupture, intraocular lens dislocation, or endophthalmitis occurred in either group during the follow-up period.

## 4. Discussion

To the best of our knowledge, this was the first study to assess the clinical outcomes of the dual-linear foot-pedal system, in which ultrasound power and vacuum are controlled separately, in cataract surgery. The advantage of this type of foot control was expected to be a more judicious use of ultrasound power and thus a reduction in the total energy dissipated in the eye. However, our results show that average phaco power and EPT values were greater in the study group than in the controls. In subgroup analysis, while the average phaco power was high in all subgroups, EPTs were comparable in the harder nucleus groups (NO5 and NO6). Importantly, ECLs did not differ between the two groups.

Novel technologies in cataract surgery have been compared to previous technologies in terms of their efficiency and safety. Helvacioglu et al. compared the OZil Intelligent Phaco torsional mode with a combined torsional/longitudinal ultrasound mode system. They found that the former, which uses a 45°-aperture angled tip, was more effective in terms of cumulative dissipated energy and fluid volume [13]. Another major innovation in cataract surgery has been the femtosecond laser. Crozafon et al. reported real-world clinical results using electronic medical data from 811 phacoemulsification cataract surgeries and 496 femtosecond laser-assisted cataract surgeries [8]. The latter were performed with significantly lower cumulative dissipated energy. Safety measures did not significantly differ between the two groups. Another study examining the safety of femtosecond laser-assisted cataract surgery based on corneal ECL and corneal thickness found no additional harm in terms of ECL [9].

During cataract surgery, the applied ultrasound energy can be detrimental to corneal endothelial cells due to both mechanical trauma from sonic waves and thermal damage [17]. The cumulative dissipated energy is therefore often used to measure efficiency and corneal ECL or corneal thickness as indicators of safety in cataract surgery. Reported ECLs range from 4% to 25% [18,19]. In the series of Conrad-Hengerer et al. ECLs at 3 months postoperatively were 8.1 ± 8.1% in a femtosecond laser-assisted group and 13.7 ± 8.4% in controls [9]. In our own study and control groups, ECLs were 12.5 ± 11.0% and 14.8 ± 12.1%, respectively, comparable to previous work. Despite the significantly larger EPT, there were no significant differences in ECLs between the two groups at 3 months postoperatively. Moreover, the percentage ECL for each nuclear opacity grade was numerically lower in the study group whereas neither corneal thickness nor corneal volume differed significantly.

The explanations for our results include the possibility that the ECC is not an absolute indicator of hazardous phaco energy levels. At high ultrasound energy levels, corneal ECL can be minimized if the corneal endothelium is avoided during the delivery of phaco energy.

Both the total phaco energy and surgical trauma, including lens material hitting the corneal endothelium, can result in ECL. Hayashi et al. suggested that mechanical contact with nuclear fragments may be the principal cause of endothelial injury [20]. Other possible causes include an increased cumulative dissipated energy, a longer aspiration time, and the use of a larger volume of balanced salt solution. Takahashi suggested that intraoperative intraocular pressure and a temperature rise of the aqueous humor should be considered as causative factors of corneal endothelial damage [21]. Another suggested factor was free radical formation by high-intensity ultrasound oscillations [21]. It was reported that hydroxyl radicals are generated, which damage corneal endothelial cells through oxidative insult. In the Venturi system, the aspiration flow rate is proportional to the vacuum [22]. Therefore, dual linear control may achieve more stable anterior chamber fluidics following an increase in phaco power by avoiding a concomitant increase in the vacuum, in turn reducing unnecessary anterior-chamber turbulence, which can be harmful to the endothelium. Our use of adaptive fluidics, newly added to the Stellairs Elite platform, may also have improved the overall performance in the study group. In a study that used a different platform, active fluidic-pressure control was shown to achieve higher surgical efficiency than a gravity fluidics configuration [23].

To minimize differences in demographic data and surgical conditions, such as pupil dilation extent and nuclear density, we studied both eyes (one as part of the study group; the other, a control) of each patient. The LOCS grade distribution was not significantly different between the two groups. A different anterior chamber depth means a different distance between the corneal endothelium and the phaco tip, which can be a confounding factor for endothelial cell loss after surgery. Therefore, we additionally analyzed the anterior chamber depth, lens thickness, and axial length. The comparison showed no significant difference between the two groups. A separate analysis was also performed using data from the other swept source OCT biometry (IOLMaster 700) and again there were no differences in all three parameters. Randomization also ensured that there were no differences in terms of the laterality of the eyes in each group. Again, the surgeon located himself superiorly to the patient and used both hands as the main manipulator for each respective eye. It is believed that this setting may reduce surgical difficulties that may occur when performing left eye surgery using the right hand. We compared the EPT between right and left eyes in each group and no differences were found. We believe that superior positioning of the surgeon has some advantages in surgery. This eliminates the need to move the phaco machine or microscope depending on the laterality of the eye. Consistent positioning of the surgeon, patient, nurse, phaco machine, and microscope would improve surgical consistency and reduce the potential for contamination or mechanical errors. We compared the corneal volume as well as specular microscopy to find differences between the two groups. Takahashi has proposed the corneal volume as a new parameter to assess the total corneal endothelial function [21].

Some limitations should be noted. Firstly, although dual-linear foot pedal control may facilitate nucleus removal, the conventional phaco mode is well developed and efficient in cataract surgery. Therefore, the EPT may not be an appropriate parameter for identifying a significant advantage of dual-linear foot control. Secondly, in this study, one experienced surgeon performed all the cataract surgeries. As the 150 cases in the study group were the surgeon’s first experience with dual-linear foot-pedal control, the learning curve associated with the unaccustomed manipulation of simultaneous pitch and yaw movements may have affected the surgical efficiency. Nonetheless, the clinical results of the study group and the conventional control group were comparable. Thirdly, the volume of fluid utilized in each surgery could not be compared due to the retrospective nature of the study. The separate control offered by the dual-linear foot pedal, including the vacuum, might result in effects on anterior-chamber fluidics that differ from those associated with conventional foot-pedal control. A measurable parameter for fluidics is the required volume of balanced salt solution, which should be compared in further studies of the two foot-pedal types.

Dual-linear control allows the surgeon to control the vacuum level and phaco power independently. This study demonstrated the safety of this method when adopted by an experienced surgeon, even during the early phase of the learning curve. The differences in ECC, CCT, CV, and BCDVA values between eyes treated with conventional foot-pedal control and those treated with dual-linear control were not significant. There were also no serious surgery-related complications in either group. Further studies, with a larger number of study eyes and a sufficient learning curve, are needed to validate the advantages of dual-linear foot-pedal control.

## Figures and Tables

**Table 1 jcm-13-00693-t001:** Patient demographics and clinical data.

Characteristic	Dual Linear Group	Control Group	*p*-Value
Age at surgery			
Mean ± SD (years)	68.9 ± 8.1		
≤60 years, n (%)	14 (14.9%)		
61–70 years, n (%)	42 (44.7%)		
71–80 years, n (%)	30 (31.9%)		
≥81 years, n (%)	8 (8.5%)		
Sex			
Males	51 (54.3%)		
Females	43 (45.7%)		
Laterality			
Right eye	43 (46%)	51 (54%)	0.688
LOCS III grade			0.206
NO2	4	6	
NO3	26	25	
NO4	34	46	
NO5	18	11	
NO6	12	6	
Baseline BCDVA			
Mean ± SD of BCDVA (LogMAR)	0.39 ± 0.40	0.35 ± 0.33	0.383
Endothelial cell count (cells/mm^2^)			
Mean ± SD	2770.2 ± 350.5	2785.9 ± 349.6	0.759
Central corneal thickness (µm)	540.0 ± 28.9	539.5 ± 28.7	0.921
Corneal volume	58.5 ± 4.9	58.7 ± 4.6	0.791
Anterior chamber depth (mm) *	3.13 ± 0.39	3.13 ± 0.39	0.992
Lens thickness (mm) *	4.58 ± 0.41	4.59 ± 0.43	0.793
Axial length (mm) *	23.45 ± 0.55	23.43 ± 0.55	0.857

BCDVA = best-corrected distance visual acuity; SD = standard deviation; * data obtained from swept source OCT (Anterion).

**Table 2 jcm-13-00693-t002:** Phacoemulsification parameters of the cataract using the dual linear and conventional (control) ultrasound methods.

Parameter	Dual Linear Group	Control Group	*p* Value
APT (s)	7.05 ± 5.31	6.82 ± 6.48	0.793
Average phaco power	28.4 ± 1.00	18.9 ± 1.74	0.001
EPT	2.05 ± 1.56	1.35 ± 1.35	<0.001

APT: Absolute phaco time, EPT: effective phaco time.

**Table 3 jcm-13-00693-t003:** Surgical outcomes in eyes treated with the two different ultrasound methods at 3-month follow-up.

Parameter	Dual Linear Group	Control Group	*p* Value
Endothelial cell loss (%)	12.5 ± 11.0	14.8 ± 12.1	0.174
Central corneal thickness (µm)	541.1 ± 29.3	541.6 ± 29.3	0.901
Corneal volume	59.1 ± 5.1	59.5 ± 4.6	0.619
Mean ± SD of BCDVA (LogMAR)	0.0 ± 0.70	0.0 ± 0.70	0.395

BCDVA = best-corrected distance visual acuity; SD = standard deviation.

**Table 4 jcm-13-00693-t004:** Changes in the number of corneal endothelial cells over time.

	Corneal Endothelial Cells (Cells/mm^2^)						
	Dual Linear Group				Control Group		
Exam	Mean ± SD	Minimum	Maximum		Mean ± SD	Minimum	Maximum
Preoperative	2770.2 ± 350.5	1811	3561		2785.9 ± 349.6	1594	3485
Postoperative							
1 week	2475.6 ± 435.4	1509	3462		2517.6 ± 411.7	1365	3328
1 month	2442.2 ± 419.3	1299	3668		2425.5 ± 452.8	1321	3679
3 months	2418.5 ± 397.4	1404	3192		2372.4 ± 448.6	1235	3372

**Table 5 jcm-13-00693-t005:** Central corneal thickness over time.

	Central Corneal Thickness (µm)						
	Dual Linear Group				Control Group		
Exam	Mean ± SD	Minimum	Maximum		Mean ± SD	Minimum	Maximum
Preoperative	540.0 ± 28.9	467	603		539.5 ± 28.7	480	617
Postoperative							
1 week	547.5 ± 30.7	474	641		551.4 ± 34.5	480	663
1 month	543.7 ± 27.9	467	608		545.1 ± 28.1	476	613
3 months	541.1 ± 29.3	474	609		541.6 ± 29.3	478	620

**Table 6 jcm-13-00693-t006:** Surgical outcomes in eyes treated using the two different ultrasound foot-pedal-control methods at 3-month follow-up, according to nuclear opacity grade.

	Nucleus Grade														
	NO2			NO3			NO4			NO5			NO6		
Parameter	Group 1 ^a^	Group 2 ^b^	*p*	Group 1	Group 2	*p*	Group 1	Group 2	*p*	Group 1	Group 2	*p*	Group 1	Group 2	*p*
APT (s)	1.60 ± 0.42	1.34 ± 0.74	0.285	3.59 ± 1.85	4.05 ± 2.20	0.534	5.97 ± 2.88	6.02 ± 2.52	0.812	8.74 ± 4.15	9.99 ± 3.16	0.281	16.89 ± 5.10	24.19 ± 14.71	0.399
Average phaco power	27.3 ± 1.7	18.3 ± 2.0	0.01	27.9 ± 1.1	18.0 ± 2.2	<0.001	28.7 ± 0.7	19.2 ± 1.5	<0.001	28.6 ± 0.8	18.9 ± 0.8	<0.001	28.7 ± 0.9	20.3 ± 0.5	<0.001
EPT	0.44 ± 0.14	0.26 ± 0.16	0.087	1.02 ± 0.54	0.74 ± 0.39	0.053	1.73 ± 0.83	1.18 ± 0.47	<0.001	2.58 ± 1.22	2.00 ± 0.67	0.276	4.94 ± 1.52	5.05 ± 3.11	0.574
ECL (%)	5.2 ± 13.5	8.1 ± 6.3	0.286	9.7 ± 11.2	11.3 ± 7.8	0.351	12.9 ± 9.3	14.0 ± 12.5	0.981	11.0 ± 9.8	21.5 ± 12.1	0.055	22.0 ± 12.1	30.6 ± 14.7	0.303
Central corneal thickness (µm)	569.3 ± 23.8	566.3 ± 20.7	0.748	537.1 ± 27.8	567.6 ± 31.4	0.970	542.4 ± 23.8	539.2 ± 28.0	0.592	539.8 ± 37.1	551.0 ± 32.9	0.220	538.7 ± 34.5	535.2 ± 22.7	0.779
Corneal volume	62.0 ± 3.3	63.2 ± 1.9	0.394	60.3 ± 3.1	60.0 ± 4.1	0.992	58.2 ± 5.8	59.5 ± 4.0	0.658	60.1 ± 5.5	59.5 ± 5.1	0.808	56.9 ± 5.7	53.2 ± 6.5	0.190

NO: nuclear opalescence, APT: absolute phaco time, EPT: effective phaco time, ECL: endothelial cell loss; ^a^ Group 1: dual linear group; ^b^ Group 2: control group.

## Data Availability

Data may be provided upon reasonable request.

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
