# Peer review of "Comparison between Early Clinical Results of Dual-Linear and Conventional Foot-Pedal Control in Phacoemulsification"

_jcm, 2024, doi:10.3390/jcm13030693_

Round 1
Reviewer 1 Report
Comments and Suggestions for Authors
Dear Authors,
Thanks for the manuscript.
Minor comments only for the methods.
- Lines 88-89: the main incision was made in the steepest axis. In view of the above it has not been used same approach superior/temporal in each case? Although unlikely, in same patients, have there been cases of one eye with temporal and the other with superior approach? Also, the steepest axis has been evaluated at IOL master 700?
- Lens thickness, axial length, and anterior chamber depth have been taken in consideration at baseline in the two groups (and subgroup analysis)? Same as in the first comment, it’s unlikely that there are differences between the two eyes, but it could maybe increase the validity of the results.
Author Response
Thank you for providing an opportunity to improve the manuscript.
- We added more explanations on the way we made the main incision. Therefore, in case one eye is WTR and the other eye is ATR, we made incision accordingly. The steep axis was determined using Toric IOL summary at 4 mm obtained from Refractive Power / Corneal Analyzer (OPD-Scan III, NIDEK, Japan). In the case where astigmatism is inconsistent and less than 0.5 D, the main incision was located at the superior temporal sclera to minimize surgically induced astigmatism.
- We compared those three data (lens thickness, axial length, and anterior chamber depth) obtained from two kinds of swept-source OCT (Anterion and IOLMaser 700) and found no differences in all three parameters. We added data from Anterion in Table 1.
Reviewer 2 Report
Comments and Suggestions for Authors
Dual-linear control allows the surgeon to control the vacuum level and phaco power independently. This study demonstrated the safety of this method when adopted by an experienced surgeon, even during the early phase of the learning curve. The differences in ECC, CCT, CV, and BCDVA values between eyes treated with conventional foot-pedal control and those treated with dual-linear control were not significant. There were also no serious surgery-related complications in either group. this was the first study to assess the clinical outcomes of the dual-linear foot-pedal system, in which ultrasound power and vacuum are controlled separately, in cataract surgery. The advantage of this type of foot control was expected to be more judicious use of ultrasound power and thus a reduction in the total energy dissipated in the eye. However, current data show that average phaco power and EPT values were greater in the study group than in the controls, possibly because the technique was newly adopted? In this paired-eye contralateral, retrospective, observational study, could the higher energy relates to doing phaco in the left eye while the surgeon is right handed?
Author Response
Thank you for providing an opportunity to improve the manuscript.
First, we are planning a follow up study after a sufficient learning curve.
Second, the surgeon was ambidextrous. The surgeon positioned himself at a superior aspect to the patient. During surgeries performed on the right eye, the phaco handpiece was manipulated with the right hand and for the left eye the phaco handpiece was manipulated with the left hand.